# Hard ImageNet: Segmentations for Objects with Strong Spurious Cues

**Mazda Moayeri**
mmoayeri@umd.edu

**Sahil Singla**
ssingla@umd.edu

**Soheil Feizi**
sfeizi@cs.umd.edu

Department of Computer Science
University of Maryland

## Abstract

Deep classifiers are known to rely on spurious features, leading to reduced generalization. The severity of this problem varies significantly by class. We identify 15 classes in ImageNet with very strong spurious cues, and collect segmentation masks for these challenging objects to form *Hard ImageNet*. Leveraging noise, saliency, and ablation based metrics, we demonstrate that models rely on spurious features in Hard ImageNet far more than in RIVAL10, an ImageNet analog to CIFAR10. We observe Hard ImageNet objects are less centered and occupy much less space in their images than RIVAL10 objects, leading to greater spurious feature reliance. Further, we use robust neural features to automatically rank our images based on the degree of spurious cues present. Comparing images with high and low rankings within a class naturally reveals the exact spurious features models rely upon, and shows classifiers suffer reduced accuracy when spurious features are absent. With Hard ImageNet's annotations and evaluation suite, the community can begin to address the problem of learning to detect challenging objects *for the right reasons*, despite the presence of strong spurious cues.

## 1 Introduction

Deep learning based image classifiers are effective but brittle to distribution shift, leading to serious issues when models are deployed to sensitive applications like medicine [7, 42]. The reliance of models on spurious features, which are predictive of class labels in training data but are irrelevant to the true labeling function, contributes to reduced distributional robustness. The lack of interpretability of deep models is a bottleneck to understanding causes and degree to which models rely on spurious features, especially on real data. Some datasets have been proposed where spurious features are manually annotated or injected into synthetic [1, 30] or application-specific [25, 21] datasets, though insights from these benchmarks may have limited transferability to more general domains.

Recently, [34] apply a mostly automated procedure for neural network interpretability [36] to discover and annotate core (i.e. essential to the class label) and spurious features at scale. [35] expand this analysis to softly segment core input regions in nearly all of the ubiquitous benchmark dataset, ImageNet [8]. Their method leverages the activation maps of neural features that highly contribute to the activation of a given class logit and detect core features for that class. However, for 15 classes, all of the inspected features were spurious, making the core-feature-based segmentation scheme infeasible, while suggesting the presence of uniquely strong spurious cues within this subset of ImageNet.

In this paper, we first gather object segmentations for these 15 classes, forming the *Hard ImageNet* dataset. We then curate an evaluation suite of ablation, noise, and saliency based analyses that leverage object segmentations to assess the degree to which models rely on spurious features. With this benchmark, we show that models rely on spurious features much more for Hard ImageNet objects

36th Conference on Neural Information Processing Systems (NeurIPS 2022) Track on Datasets and Benchmarks.

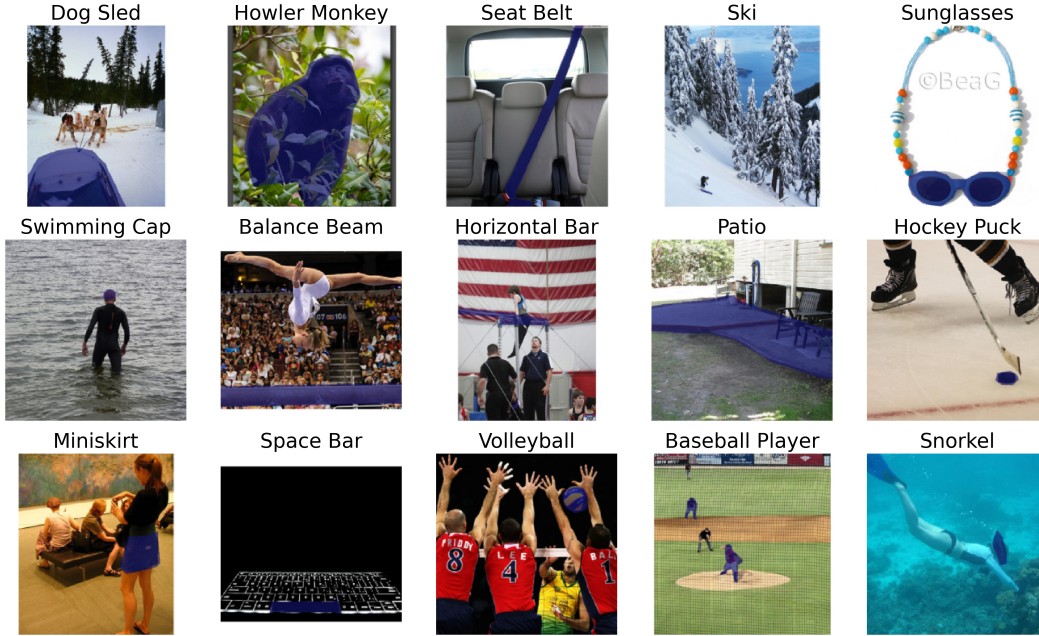

Figure 1: *Hard ImageNet* classes. Object segmentation masks are overlaid in purple.

than more typical objects, as represented by the RIVAL10 dataset [26], who's classes appear in multiple standard benchmark datasets (ImageNet, CIFAR10 and STL10 [22, 5]). We then investigate properties of Hard ImageNet objects, observing that they are significantly smaller and less centered, likely leading to models favoring spurious features. Further, we complement our instance-wise object segmentations by ranking each image within a class based on the strength of spurious cues present, leading to clearer visualizations of the specific spurious features involved. The image rankings also open the door to last-layer retraining of models on data where the presence and absence of spurious features is balanced, an efficient new technique for reducing spurious feature reliance [20].

With Hard ImageNet, we go beyond single-label annotations to shed new insights on the data and model conditions that lead to increased spurious feature reliance in deep image classifiers. We hope our evaluation suite and challenging dataset can inspire new and improved training procedures that result in models that classify *for the right reasons*, towards more reliable and trustworthy AI.

## 2 Related work

The tendency of models to rely on spurious features due to simplicity bias [32] or short-cut learning [11] has garnered much attention, due to its consequences on fairness and generalization, leading to the proposal of many algorithms to mitigate spurious dependencies [30, 17, 23, 29, 28, 27]. However, in comprehensive evaluations, the gains of these methods over empirical risk minimization are often marginal or inconsistent [41, 15]. Recently, a simpler approach for reducing spurious feature reliance has been proposed, where only a final linear layer is retrained over data that balances the presence of spurious cues, thus encouraging models to downweight spurious features, which are less predictive of class labels in the balanced set [20]. A bottleneck to this approach, along with many of the other mentioned algorithms, is that they require knowledge of the degree to which spurious cues are present in each sample; the image rankings of our dataset may address this need.

Several datasets have been created to aid in studying spurious feature use, including many with synthetically-injected spurious correlations [30, 1, 12]. A noteworthy example is ImageNet-9, which swaps backgrounds across classes to assess background sensitivity, using analyses similar to our ablation studies [40]. While insightful, the use of simple classification tasks and unnatural data in these synthetic sets may limit their impact in practice. A number of real-world datasets have also been proposed [21, 14, 19, 25], though they are usually very specific to the domain of their application.

Recent work appeals to additional annotations to assess model sensitivity to various input regions via noise-based analyses [26, 37]. Notably, [35] annotate core and spurious input regions for nearly all of ImageNet in a mostly automated procedure, leveraging neural feature visualizations [36] and the improved interpretability of an adversarially robust network [39], resulting in *Salient ImageNet-1M*, a large-scale, natural, and general dataset for evaluating and improving model reliance on spurious features. Namely, they used activation maps of robust neural features corresponding to core features as soft segmentations of core regions (see Appendix C). However, for 15 classes, all neural features inspected were spurious, making their segmentation framework inapplicable. Arguably, these classes are the most interesting, as the underlying robust network used for segmentation relied on spurious features the most for these classes, indicating that this subset of data contains the strongest spurious cues. We manually collect object segmentations for these 15 classes, design a suite of evaluation metrics, and study the properties of this data that leads to heightened spurious feature reliance.

## 3   Hard ImageNet Dataset

### 3.1   Overview

Our dataset, *Hard ImageNet*, consists of images from 15 ImageNet [8] synsets. These classes are *Dog Sled, Howler Monkey, Seat Belt, Ski, Sunglasses, Swimming Cap, Balance Beam, Gymnastic Horizontal Bar, Patio, Hockey Puck, Miniskirt, Keyboard Space Bar, Volleyball, Baseball Player,* and *Snorkel*, selected because for each class, all features annotated in [35] were spurious with respect to the class. Specifically, [35] inspect the five neural features (i.e. nodes in the penultimate layer) of an $\ell_2$ adversarially trained ResNet50 per class that contribute most to the activation of the corresponding logit. Each class-feature pair is annotated as *core* or *spurious*, based on if the detected feature is essential to the class label. Over all classes, the vast majority of class-feature pairs inspected were deemed *core*. Thus, having all five most important features serving *spurious* roles suggests that each Hard ImageNet class uniquely has very strong spurious cues that models favor over core features.

We collect object segmentation masks for these 15 classes, resulting in a dataset with $19,097$ training samples and $750$ validation samples. We maintain the same train/validation split as ImageNet. Figure 1 visualizes an example and its object segmentation for each Hard ImageNet class. Additionally, we provide class-wise rankings for each image in the Hard ImageNet training set based on the strength of spurious cues present in the image. We base these rankings on the activations of the annotated neural features for each class (see Section 6 for more details).

**Impact** First, Hard ImageNet segmentations complete the *Salient ImageNet* dataset, which allows for the exploration of going beyond single-label supervision in training ImageNet models to predict *for the right reasons*. That is, with object segmentations available during training, models can be guided to rely more on core features than spurious ones, leading to improved generalization to domains where spurious correlations are broken. Moreover, Hard ImageNet annotations shed insight on the conditions for models and data under which spurious features are favored: through our suite of evaluation metrics (see Section 4), the effect of training procedures and model architectures on spurious feature reliance can be compared, while analysis of segmentation masks reveal how the shape, size, and location of objects in images affects model sensitivity to the object (see Section 5).

### 3.2   Collection Procedure

We collect Hard ImageNet object segmentations over Amazon Mechanical Turk. To ensure quality, annotations are collected in many phases. First, workers must pass a *qualification exam*, consisting of segmenting one sample from each class. All workers take the same qualification exam, and only workers who achieve an intersection-over-union (IoU) score of at least $0.65$ pass. For reference, any IoU greater than $0.5$ is considered successful in the Pascal VOC object detection evaluation [9]. Workers meeting this bar demonstrate proficiency with the Mechanical Turk platform and sufficient understanding of the relevant objects to be segmented. Then, workers sign a *informative consent form*, in which we explain the nature of and purpose for their work, and also share answers to the qualification exam and tips for improved segmentations so to correct for common errors. Finally, qualified and consenting workers move on to the full data collection stage, where we release batches in rounds, monitoring quality with *attention checks* (randomly inserted images with ground truth segmentations) and updating class-specific tips based on errors observed on attention checks.

We place attention checks with a frequency of $10\%$ for the validation split and $5\%$ for the training split. The average IoU score on attention checks was $0.76$. Workers with very low quality on attention checks were removed from the study, and their segmentations were redone in later rounds. Also, each image in the validation set is annotated five times, with annotations consolidated via a pixel-wise majority vote to correct for errors in any individual segmentation. Figure 2 shows the IoU scores of each validation set segmentation to the consolidated final validation masks. The vast majority of segmentations are highly agreed upon, with a small minority having near zero agreement, indicating that the extra rounds of annotation had a meaningful corrective impact.

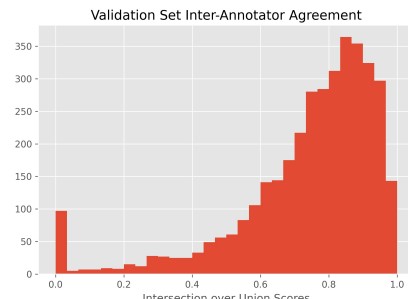

Figure 2: Agreement of individual segmentations with final averaged segmentation for Hard ImageNet validation split.

Throughout our data collection, we strive to create a transparent and collaborative working environment. In the consent/information stage, we invite feedback and explicitly encourage our workers to directly contact us with questions. Moreover, we reward workers with fair salaries (estimated to be $12 to $16 an hour depending on annotation speed) and bonuses (between $5\%$ and $7\%$ depending on number of annotations) for continued work. We also invite workers who previously partook in RIVAL10 [26] annotation, as their experience from a similar task leads to improved annotation quality, and rewarding loyalty improves worker well being.

## 4 Benchmarking Spurious Feature Reliance

We compile a suite of evaluation metrics that use object segmentations for assessing image classifier reliance on spurious features. We focus this section on demonstrating the significantly higher degree of spurious feature reliance for models on Hard ImageNet compared to other more typical data. As a baseline, we use RIVAL10 [26], which is comprised of images from 20 ImageNet synsets, selected so that each synset corresponds to a class from CIFAR10 [22] (i.e. two synsets per CIFAR10 class). Importantly, RIVAL10 includes an object segmentation per image, collected in a similar fashion to Hard ImageNet's annotations. We consider RIVAL10 data to be more typical of what is commonly studied in image classification, as many or all of the classes also appear in the standard benchmark datasets CIFAR10 and STL10. We note that we use samples in RIVAL10 that were originally in ImageNet's validation split, as opposed to using RIVAL10's own train/test split, which leaks ImageNet training samples to their test split. For Hard ImageNet, we also evaluate over images originally in ImageNet's validation split, which corresponds to Hard ImageNet's validation split.

Across our evaluations, we use a convolutional neural network (ResNet50 [13]) and a vision transformer (small DeiT [38]) of roughly equal size, both pretrained on ImageNet. To evaluate these models 'off the shelf', we map samples from Hard ImageNet and RIVAL10 to their original ImageNet class indices. Thus, for RIVAL10 evaluation off the shelf, we actually consider 20 target classes, and so we refer to this as RIVAL20. We also include finetuned versions where a final linear layer is trained for each model atop representations from the corresponding fixed feature encoder. For RIVAL10, we finetune for both 10-way and 20-way classification. While the emphasis of this section is *data*, in practice our evaluation metrics are intended for comparison of *models* of diverse types.

### 4.1 Ablation

Object segmentation masks allow for removal of the object corresponding to the class label for any sample. Classifying an ablated image to its original class indicates that the model uses background information as it does not require the object's presence to predict its class. While these ablated images are out of distribution, accuracy drop due to ablation can still inform the degree to which the model relies on spurious background features. We consider three types of ablation. First, we replace all object pixels with $0.5$, graying out the object. Second, we replace all pixels in the *bounding box* of the object with gray, thus also removing object shape information. Finally, we replace the bounding box of the object with a tile adjacent to the object in the image. In cases where a tile is smaller than the bounding box, we repeat the tile to fill the region, as in [40]. Note that all ablations replace each

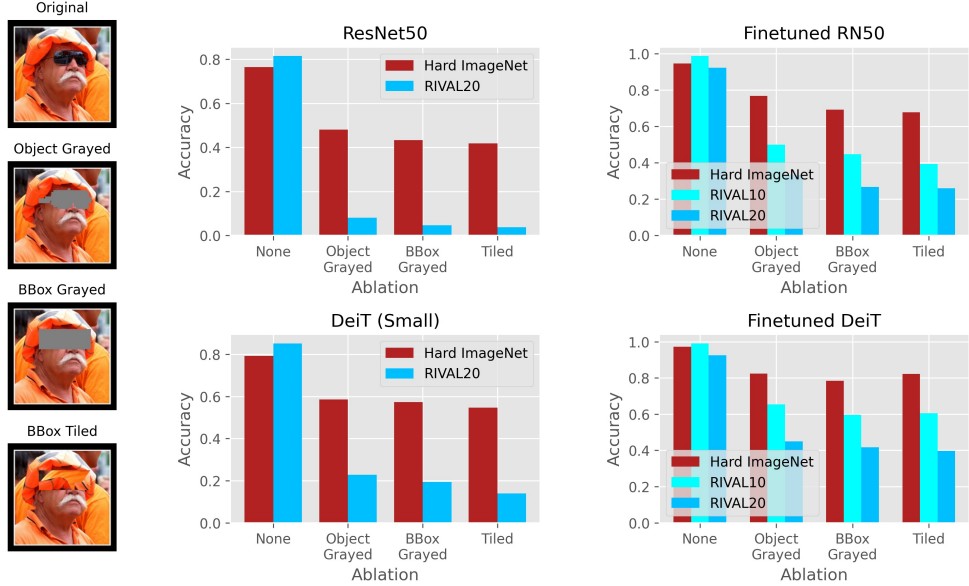

Figure 3: Accuracy under ablation. Accuracy drops much less when Hard ImageNet objects are ablated than when RIVAL10 objects are ablated. Example ablations for class *sunglasses* shown left.

instance of an object in an image separately (i.e. we consider multiple boxes bounding each instance as opposed to one box containing all instances) so to retain as much background as possible.

Figure 3 shows that ablation leads to a much smaller accuracy drop for Hard ImageNet objects than for RIVAL10 objects, in both the finetuned and off the shelf case, and for all types of ablation. In fact, for an off the shelf DeiT (i.e. performing 1000-way classification), Hard ImageNet images are predicted to their class over $50\%$ of the time under all ablations, despite the fact that the object essential to the class label is actually missing, rendering the predictions inaccurate (see Appendix D). This is several times larger than the rate for RIVAL20. In finetuned models, accuracy under ablation is still significantly larger for Hard ImageNet, though this gap is smaller. We note that random guessing yields much higher accuracy in these cases because the number of classes is multiple orders of magnitude smaller. Because number of classes varies across the three datasets considered, direct comparison is more challenging, but Hard ImageNet accuracy is still substantially larger than RIVAL10 ablated accuracy, where random guessing achieves $5\%$ higher accuracy than on Hard ImageNet.

## 4.2 Relative Foreground Sensitivity

We now turn to the noise-based metric *relative foreground sensitivity* ($RFS$), introduced with RIVAL10 [26]. To compute $RFS$, equal amounts of Gaussian noise is added to foreground and background regions. Then, the gap in accuracy drops (i.e. how much more accuracy drops due to foreground noise than background noise) is normalized to allow for comparison across models with varying noise robustness, as derived extensively in [26] and [35]. Higher $RFS$ scores entail greater sensitivity to noise in foregrounds relative to backgrounds. We consider adding both $\ell_\infty$ and $\ell_2$ noise, where in the latter case, we fix the $\ell_2$ norm of noise added in both backgrounds and foregrounds. These types of noise have opposite size biases, as $\ell_2$ noise results in a higher $\ell_\infty$ norm of noise is smaller regions, while $\ell_\infty$ noise results in a higher $\ell_2$ norm of noise in larger regions.

In figure 4, we see that for all models and across noise levels, $RFS$ is substantially lower when evaluated on Hard ImageNet than on RIVAL10 or RIVAL20. In fact, $RFS$ **is negative in many cases**, indicating that noise in the background reduces accuracy more than noise in the foreground. For finetuned models at low levels of $\ell_2$ normalized noise, $RFS$ is comparable across datasets, but otherwise, models appear to be significantly more sensitive to noise in backgrounds than they are to the actual object in Hard ImageNet, suggesting that models use object information far less than surrounding context. Arguably, despite achieving high accuracy, this result suggests that models are not actually learning to detect Hard ImageNet objects, instead relying on spurious background cues.

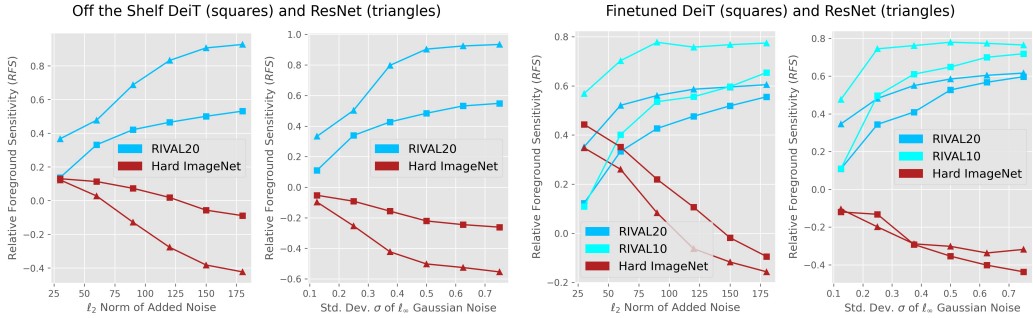

Figure 4: Relative Foreground Sensitivity ($RFS$) for DeiT and ResNet, off the shelf (left) and finetuned (right). $RFS$ for Hard ImageNet is significantly lower, and often negative, indicating greater sensitivity to background noise than foreground noise.

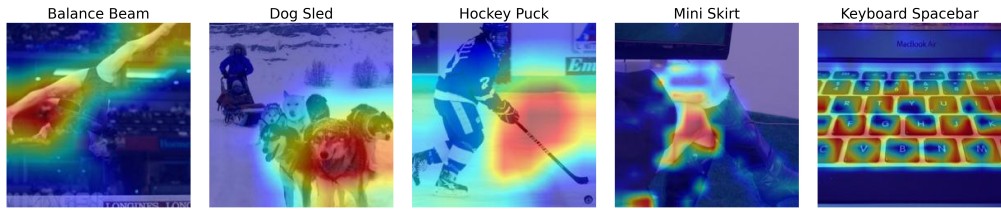

Figure 5: GradCAMs of Hard ImageNet instances with poor saliency alignment for finetuned ResNet50 (left three) and finetuned DeiT (right two). Spurious features of *gymnast, dogs, hockey stick and ice, legs* and *keys* for each class respectively are highlighted.

## 4.3 Saliency Alignment

Our third metric makes use of the interpretability method GradCAM [31], which assigns a saliency score to each input pixel, proxying its importance to the prediction. While GradCAMs are qualitative, we perform a quantitative analysis by computing the alignment of GradCAMs to objects. Specifically, we compute IoU of object segmentations and GradCAMs (binarized with a threshold of $0.5$) as in [26]. Figure 6 shows IoU scores across models are significantly lower for Hard ImageNet objects than RIVAL10 objects, indicating that backgrounds are more salient to Hard ImageNet classification.

With saliency alignment, we can also automatically extract instances with low IoU scores to reveal specific spurious background cues. Some examples are visualized in figure 5. Here, we see that other features present in the image but non-essential to the class label are arguably more prominent and

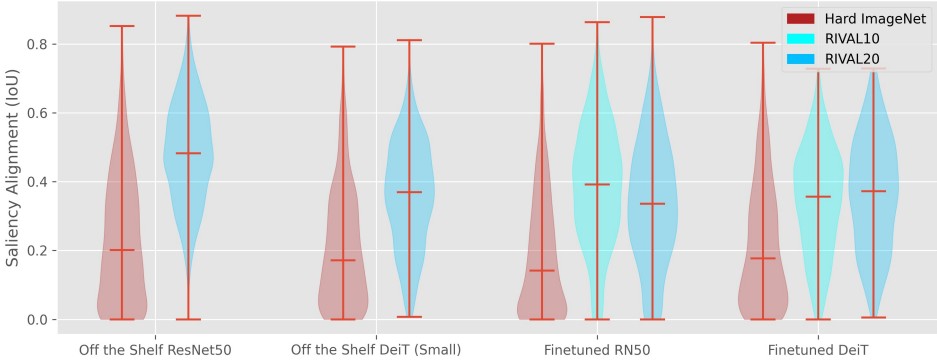

Figure 6: Alignment of GradCAM saliencies to object masks. Low alignment indicates more saliency is placed in the background. Across all models, saliency alignment is lowest for Hard ImageNet.

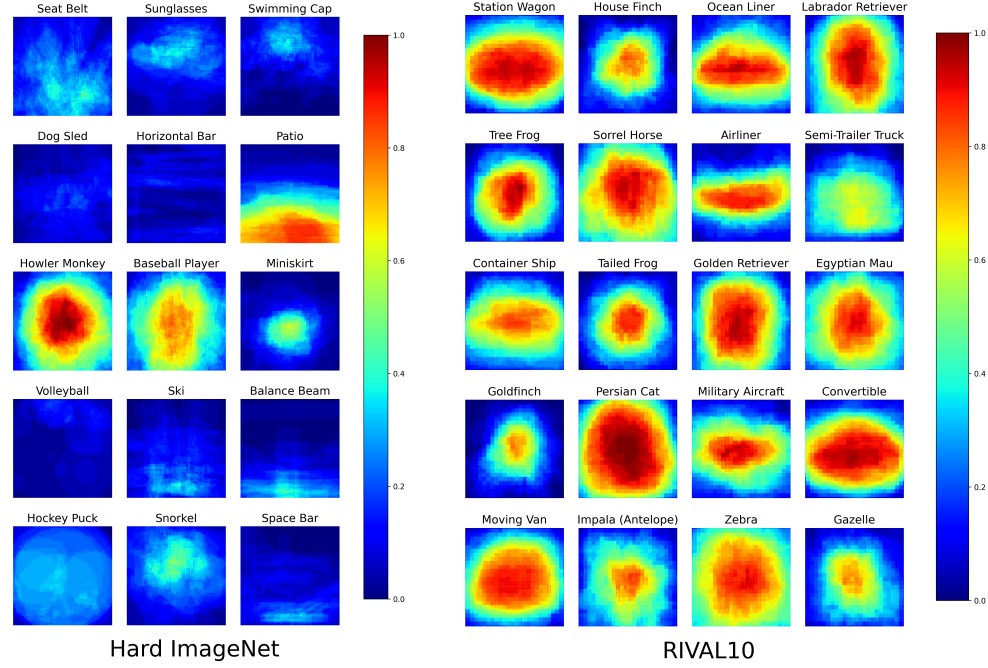

Figure 7: Average object position for Hard ImageNet and RIVAL10. Hard ImageNet objects occupy less space and are often not centered, unlike RIVAL10 (and most other ImageNet) objects.

easier to recognize than the actual object. For example, the keys on a keyboard are more noticeable than the spacebar below, as they take up more of the image and contain more detail than the uniformly colored spacebar. Further, the location of the objects in these examples makes them appear as though they are in the background, as they reside along the edges of the image, as opposed to the center. We investigate the size and location of Hard ImageNet objects in more detail in the following section.

## 5 Hard ImageNet Objects are Smaller and Less Centered

Figure 8 shows that compared to RIVAL10, Hard ImageNet objects occupy substantially less space both in their original images and after the standard ImageNet test time augmentation (consisting of resizing and center cropping). Some reasons for this are that Hard ImageNet objects are i. simply smaller (*hockey puck, swimming cap*), ii. oddly shaped so that square crops include lots of background (*spacebar, ski*), iii. often co-occurring with much larger spurious features (*snorkel, balance beam*).

Interestingly, the standard test time augmentation only slightly increases the percent of the image the object occupies, unlike for RIVAL10 objects, which see a larger increase. The lack of a center bias in object locations likely contributes to the reduced benefit of augmentation, as a center crop would not amplify non-centered objects. In fact, we find that this augmentation often removes large portions of Hard ImageNet objects, as shown in Figure 9. While in the worst case at most 25% of a RIVAL10 object is lost due to the standard augmentation, much more of the object is lost in Hard ImageNet, including cases where **standard augmentation completely removes Hard ImageNet objects**.

We visualize the class-wise average of object segmentations after standard augmentation in figure 7. Indeed, Hard ImageNet objects are smaller and less centered than RIVAL10 objects. For *howler monkey* and *baseball player*, which are relatively larger and more centered, we conjecture that the increased spurious feature reliance is due to easily recognizable spurious cues. Namely, howler monkeys are usually photographed in dense foliage with many leaves and branches. For baseball players, the uniform green grass and brown dirt of the baseball field is likely very easy for a deep network to detect. While patios are not centered, they do take up lots of space, but they often co-occur with patio furniture, which models may rely on more so than the patio itself. Finally, we note the smaller size

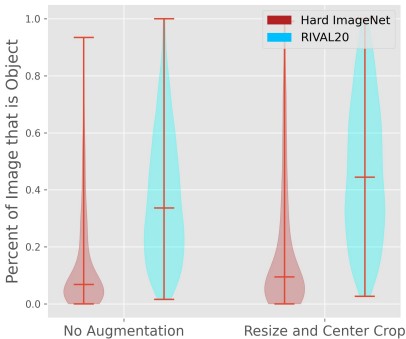
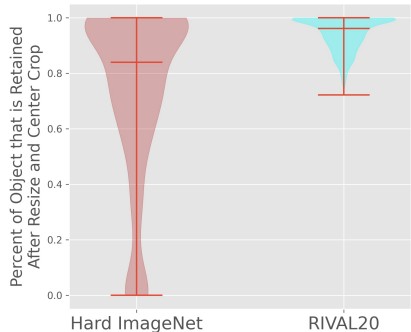

Figure 8: Hard ImageNet objects occupy less space in images than RIVAL10 objects both before and after standard augmentation.

Figure 9: Hard ImageNet objects are often substantially cropped out by the standard ImageNet test time augmentation.

of the finches in RIVAL10; RIVAL10's original analysis observed greater spurious feature reliance in the bird class, further suggesting that small object size leads to greater model use of backgrounds.

## 6 Ranking Images by Spurious Feature Presence

To complement Hard ImageNet segmentation masks, we annotate each image in the training set with a class-wise rank corresponding to the strength of spurious features present in the image. We determine these ranks by appealing to neural features (nodes in the penultimate layer) of the $\ell_2$ adversarially trained ResNet50 used to construct the Salient ImageNet dataset [34, 35]. Per class, we first rank all images by feature value on each of the five annotated features (which are all spurious for Hard ImageNet classes). We then sort the average of these ranks to obtain our final image rankings.

Figure 10 shows the highest and lowest ranked images for classes *ski, snorkel*, and *howler monkey*. Samples with low spurious rank are classified with significantly lower accuracy. Also, viewing the high and low ranked images reveals the strong spurious cues present in Hard ImageNet. For example, models appear to rely on *snow* and *skiers* to classify *skis*. Similarly, the absence of *people* and *water* for *snorkels* and *trees* for *howler monkey* reduces spurious ranking and classification accuracy. Viewpoint may also be a spurious feature for *howler monkey*, as highly ranked images are all taken from below, with the sky and trees as the backdrop. Viewpoint, along with backgrounds, was observed to be a spurious correlation that contributes to large accuracy drop when broken in ObjectNet [2].

While image ranking facilitates the interpretation of spurious features naturally (i.e. without using visualizations that make artificial changes to the image, which were observed to be less effective in [4]), they can also be used to reduce spurious feature reliance. Namely, retraining a linear layer atop fixed encoders on data that balances the presence and absence of spurious features has been found to greatly improve performance on instances where spurious correlations are broken [20]. Spurious image ranking allows for the construction of these balanced datasets, explored in the following section.

## 7 Improving Models with Hard ImageNet Annotations

We now explore baseline methods to harness Hard ImageNet's annotations for improved model classification (i.e. with reduced reliance on spurious features). We focus our study on finetuned models pretrained on ImageNet, using ResNet50 and DeiT (Small) as in Section 4. We keep features fixed during finetuning, only optimizing the parameters of a new final layer for the 15-way Hard ImageNet classification.

We employ two approaches for mitigating spurious feature reliance. [35] propose **Core Risk Minimization** (CoRM) as an alternative to ERM when segmentations of core (i.e. not spurious) regions are available; Hard ImageNet's object segmentations fulfill this prerequisite. Specifically, the objective of CoRM is to minimize classification error over the distribution of images *with noise*

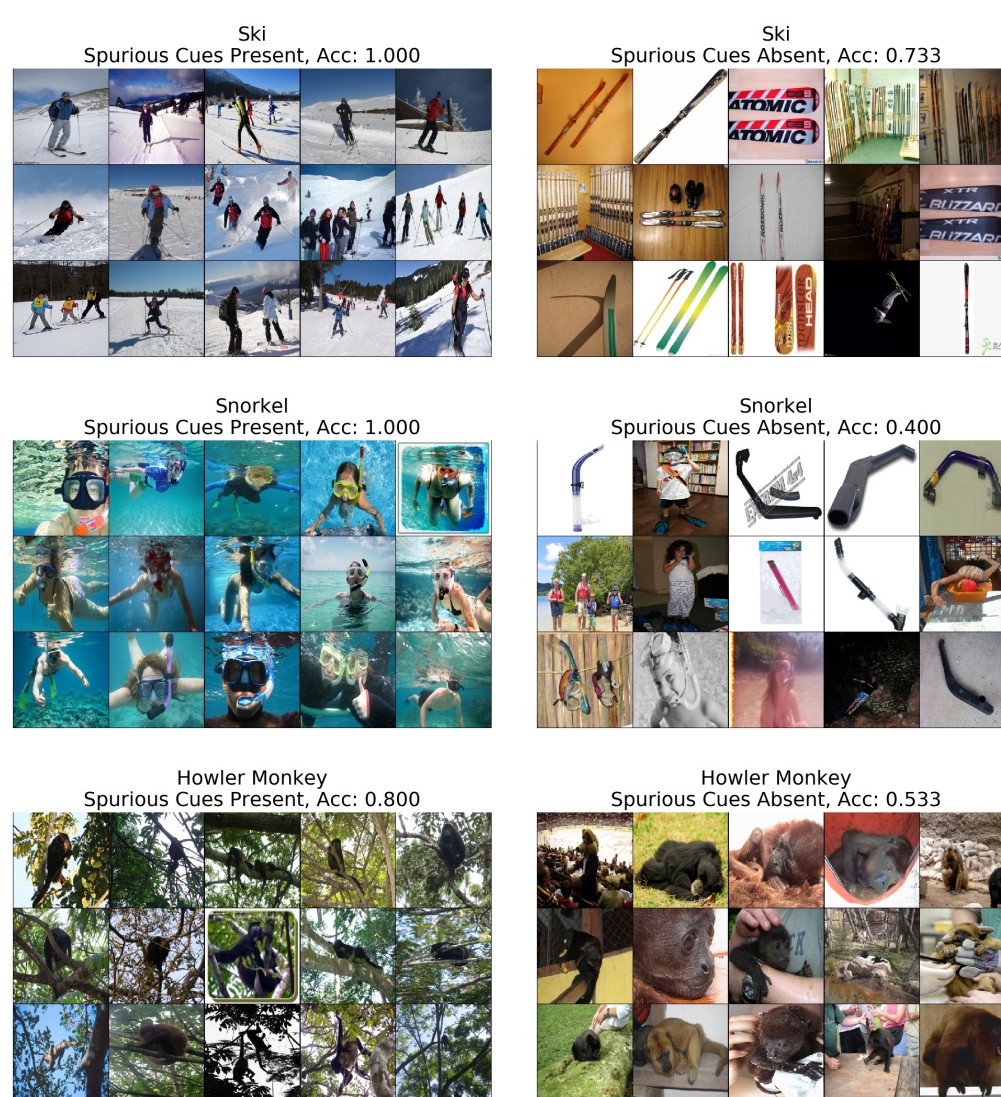

Figure 10: Examples of high (left) and low (right) spurious feature presence, based on Hard ImageNet rankings. Accuracies shown are for a standardly trained ResNet50.

*applied to non-core regions*, so that the optimal classifier predicts correctly even when spurious features are corrupted. In that work, *random noising*, where small amounts of Gaussian noise are added to non-core regions with probability $p = 0.5$, and *saliency regularization*, where the $\ell_2$ norm of the gradient on non-core pixels is added to the classification loss, were applied in tandem to improve relative core sensitivity (an analagous metric to $RFS$). [20] propose **Deep Feature Reweighting** (DFR), in which retraining a final linear layer using a *balanced dataset* reduces spurious feature reliance. The balanced dataset consists of a subset of the training data containing an equal portion of samples with and samples without spurious features, essentially breaking spurious correlations that impede generalization to minority groups. Using Hard ImageNet's image rankings, we extract the top and bottom 100 images for each class to form the spurious-balanced subset.

Table 1 shows that these two methods can considerably reduce model reliance on spurious features, improving numbers across all metrics in our benchmark. Between the two approaches, CoRM appears to lead to more improvement in saliency alignment and $RFS$, while DFR yields beter results for ablation. Combining CoRM and DFR leads to even better performance with respect to accuracies under ablation. While improvements are at times small, we note that in these experiments, the vast majority of model parameters are left unchanged, as we only train a new final layer. We leave the

| Method | | Ablation Accuracies (↓) | | | | $RFS$ (↑) | | Saliency (↑) |
|---|---|---|---|---|---|---|---|---|
| CoRM | DFR | None (↑) | Gray | Gray BBox | Tile | $\sigma = 0.25$ | $\sigma = 0.5$ | IoU |
| | | | | | Finetuned DeiT (Small) | | | |
| ✗ | ✗ | **96.79** | 84.22 | 80.48 | 81.15 | $-0.19$ | $-0.35$ | 20.90 |
| ✓ | ✗ | 96.39 | **81.02** | 78.74 | 80.75 | **0.02** | **−0.19** | 21.57 |
| ✗ | ✓ | 96.66 | 81.28 | **77.01** | 77.94 | $-0.20$ | $-0.33$ | 21.63 |
| ✓ | ✓ | 96.52 | 82.35 | **77.01** | **77.81** | $-0.10$ | $-0.29$ | **21.99** |
| | | | | | Finetuned ResNet50 | | | |
| ✗ | ✗ | 94.25 | 75.94 | 69.39 | 67.38 | $-0.18$ | **−0.27** | 18.44 |
| ✓ | ✗ | 92.91 | 76.20 | 69.12 | 68.32 | **−0.08** | **−0.27** | **20.43** |
| ✗ | ✓ | **94.39** | 73.53 | 67.51 | 66.71 | $-0.27$ | $-0.35$ | 18.39 |
| ✓ | ✓ | 91.31 | **72.59** | **63.64** | **63.90** | $-0.23$ | $-0.31$ | 20.35 |

Table 1: Final layer retraining improves faithful learning on Hard ImageNet. Results shown for entire benchmark under two different training approaches: i) Core Risk Minimization (CoRM) via *random background noising* and *saliency regularization*, and ii) deep feature reweighting (DFR) using a *spurious-balanced training subset*. We also report results for the combination of the two approaches and ordinary finetuning (as a baseline) under two architectures. Relative Foreground Sensitivity ($RFS$) is evaluated under two $\ell_\infty$ noise levels, indicated by $\sigma$. Saliency refers to *saliency alignment* as measured by intersection over union (IoU).

door open to new approaches for improving the *faithful* learning of Hard ImageNet objects, including training models from scratch.

## 8 Discussion

Our benchmark brings a new perspective to classification, as we not only seek models that predict accurately, but also predict for the right reasons. Assessing model performance using accuracy alone can obscure key misconceptions held by models, which may only become apparent when models are deployed to new domains at test time. Moreover, design decisions such as training strategy and architecture may affect the degree to which spurious features are relied upon, as observed in [26]; this dataset and accompanying benchmark can reveal these model differences. Finally, we emphasize the need to understand model behavior under "bad" data; that is, images where the object of interest is not centered or large, unlike most cases. With models becoming increasingly data hungry, it is inevitable that some portion of the data will not capture objects in ideal conditions. Further, certain objects simply are not well suited to be captured prominently (i.e. large and centered) in square photos. Figuring out how to learn to recognize objects from these suboptimal data conditions will be an important challenge to extend the impressive performance of deep classifiers from standard datasets to many more realistic settings.

With Hard ImageNet, the community can evaluate the capacity of any ImageNet trained model to faithfully learn challenging objects, and also explore how going beyond single class label annotations can lead to improved image classifiers. While segmentation masks are expensive to collect, procedures that are much more automated already exist [35], and we envision newer ones are likely to emerge with time. Also, the procedure with which we ranked images was largely automated, indicating that these types of annotations are by no means prohibitively expensive. We hope Hard ImageNet can lead to new perspectives on both training and evaluation paradigms for image classification.

## 9 Acknowledgements

This project was supported in part by NSF CAREER AWARD 1942230, HR001119S0026 (GARD), ONR YIP award N00014-22-1-2271, Army Grant No. W911NF2120076, the NSF award CCF2212458, an AWS Machine Learning Research Award.

