# OpenReview forum: "Hard ImageNet: Segmentations for Objects with Strong Spurious Cues"
_NeurIPS.cc/2022/Track/Datasets_and_Benchmarks — NeurIPS 2022 Datasets and Benchmarks _

### Official Review · Reviewer_TYq1 · 2022-07-24
**Nice dataset about spurious Cues**

**Rating:** 7
**Confidence:** 4

**Strengths:**

1. The challenge in learning with strong spurious cues is a important topic to tackle
2. The dataset annotation collection is convincing and of high quality, the sufficient training set size would also enable developing better training methods
3.  Comprehensive benchmarks metrics and experiments are conducted to compared different models, the results are convincing;
4. A analysis of the segmentation masks to reveal how the shape, size and location influence the sensitivity;
5.  Class-wise rankings for each image in the Hard ImageNet training set based on the strength  of spurious cues are provided, which can be useful for analysis and developing more advanced methods.

**Weaknesses:**

The author mentioned, "While these ablated images are out of distribution, accuracy drop due to ablation can still inform the degree to which the model relies on spurious background features". However, it has been pointed out in [1] that " Without re-training, it is unclear whether the degradation in model performance comes from the distribution shift or because the features that were removed are truly informative". In their experiment, they found " It turns out that removing information is quite hard. With 90% of the inputs removed the network still achieves 63.53% accuracy compared to 76.68% on clean data. This implies that a strong performance degradation without re-training might be caused by a shift in distribution instead of removal of information. " Though the author used the finetuned version via a lightweighted last-layer training, it would be interesting to see whether the gap narrows down when re-training is conducted as in [1]. Don't take me wrong, I think the finetuning setting is still interesting, it certainly reflects that the Spurious information are easier to be used by the model to some extent. Nevertheless, re-training can provide additional information


It would be better if the main paper and the appendix can be reorganized so that the main paper itself is in a self-contained manner. It appears to me that some important content that should be included in the main paper is put into the appendix.





[1]Hooker, Sara, et al. "A benchmark for interpretability methods in deep neural networks." Advances in neural information processing systems 32 (2019).

**Additional Feedback:**

So the comments above.

**Clarity:**

This paper is easy to follow, though I would suggest a reorganization of the main paper and appendix. It occurs to me that some important content is put in the appendix, which makes the main paper not quite self-contained.

**Correctness:**

The claims look fine to me. The description of data collection is complete and detailed. Experimental details and evaluation metrics are also well documented.

**Documentation:**

The details on data collection and organization are sufficient.

**Ethics:**

Though there are no severe ethics issues from what I can see, it would be better if the author can have a separate brief section describing such aspects.

**Relation To Prior Work:**

The description of the comparison to related work are sufficient .

**Summary And Contributions:**

In general, this is an interesting paper that tackles the challenge in learning with strong spurious cues. It is known that deep classifiers can rely on spurious features, which jeopardize the generalization ability. A well-collected dataset is necessary for formally tackling the problem. This paper identifies 15 classes in ImageNet with strong spurious cues and collect high-quality segmentation map for the images. The dataset not only contains images for validation. With 19097 images for training, it further opens the venue for more advanced methods that can leverage such information during training to reduce reliance on spurious cues. The author provides a detailed description of data collection, and benchmarking metrics and the results are solid and convincing. The author also shed some light on the characteristic of the hard ImageNet objects (i.e. smaller and less centered). To this end, ranking images by spurious feature presence are conducted and further used in the methods described in the appendix to achieve improved performance.

---

> ### Author Response · Authors · 2022-08-12
> **Thanks! Paper Reorganization to come with extra page if paper is accepted; Exploring ROAR (remove and retrain)**
>
> We thank the reviewer for all their feedback.
>
> - **On removing and retraining**: We opted for the simple ablation analysis as opposed to a removal and retraining approach in order to reduce the computational overhead needed to use our benchmark. Also, prior work has identified that in the process of removing and retraining, the new model may learn completely different features in order to make up for lost accuracy. Thus, while accuracy loss after removal and retraining gives some notion of how useful the removed features were, the signal may be misleading, as the predictivity of the new features that model relies upon may be independent of the predictivity of the features removed. In other words, even if a model can predict from the background alone, it does not entail that the original model is only using the background.
> Nonetheless, we finetune pretrained models on foreground ablated images for Hard ImageNet, RIVAL10, and RIVAL20. We find that even after removing and retraining,  Hard ImageNet classification has the highest accuracy by far, corroborating our simple ablation results. We present these results in the table below. Recall that direct comparison across datasets is complicated due to a difference in number of classes. However, even when comparing Hard ImageNet to the easier (w.r.t number of classes) dataset RIVAL10, models still attain substantially higher accuracy under ablation for Hard ImageNet.
>
> |             | Hard ImageNet | RIVAL20       |  RIVAL10     |
> | :---        |    :----:     |    :----:     |         ---: |
> | ResNet      | 87.9          | 55.9          |75.3          |
> | DeiT        | 91.2          | 64.1          |83.8          |
> Accuracy on foreground-ablated data for models finetuned on foreground-ablated data.
>
> Further, we note that the distribution shift incurred via ablation is roughly equivalent for both Hard ImageNet and RIVAL10/20. Thus, performance degradation cannot be exclusively due to the distribution shift, as the amount of degradation is significantly higher for RIVAL10 /20, where we claim models rely on foregrounds much more.
>
> - **Reorganization of paper**: If our paper is accepted, we will make use of the additional content page to move up our analysis on improving models using Hard ImageNet annotations as requested. We thank the reviewer for reading through the appendix to see this section, and concur that it would be nice to include in the main text (space permitting).
>
> Thank you again for all your comments and useful discussion.

---

> > ### Author Response · Authors · 2022-08-12
> > **citation for mentioned prior work**
> >
> > Apologies for leaving off this citation re ROAR:  Ismail, A.A., Bravo, H.C., & Feizi, S. (2021). Improving Deep Learning Interpretability by Saliency Guided Training. NeurIPS.
> >
> > Section 5 in this paper discusses how data redundancy can lead models to achieve very high accuracy after ROAR, despite the original model still potentially relying on the removed features.

---

### Official Review · Reviewer_M3X2 · 2022-07-25
**Dataset to evaluate model predictions on examples with strong spurious cues**

**Rating:** 6
**Confidence:** 2

**Strengths:**

The paper focuses on an interesting problem consisting in better understanding the predictions made by a model and proposes to annotate a part of a classical vision dataset to facilitate research on this topic.

**Weaknesses:**

The weaknesses of this work intersect for me with the points listed in section "Correctness"

**Additional Feedback:**

N/A

**Clarity:**

The paper is well written. A conclusion section would have been a nice addition.

**Correctness:**

- I have some reservations about my ability to see if the dataset is built in a sound way. Indeed, the selection of images to annotate is based on a previous work [28] available only on arxiv. This criterion is from what I understand the only criterion taken into account to define which are examples with very strong spurious clues.
- Regarding the suite of evaluation metrics, the authors state l.173 that "Hard ImageNet images are predicted to their class over 50% of the time under all ablations, despite the fact that the object essential to the class label is actually missing, rendering the predictions incorrect". My impression is that there are not enough arguments to justify that the prediction should be considered incorrect. Indeed, by combining the description made of the boxes with the images given in Figure 1, I have the intuition that for some images given the limited possibility of classes that can be predicted the classification would not necessarily be wrong. In other words, what would have been the correct classification in the case where the space bar is hidden on the 2nd figure from the left in the last row of Figure 1?

**Documentation:**

- In the paper and on the open review page the authors indicate well the license of the model (but this last one is not recalled on the home page of the dataset [https://mmoayeri.github.io/HardImageNet](https://mmoayeri.github.io/HardImageNet)), explains well data collection and indicates who is the contact point concerning the dataset.
- Concerning the usability of the dataset and the evaluation metrics, I think that a guide for the user arriving on the GitHub repository - [https://github.com/mmoayeri/hardImageNet](https://github.com/mmoayeri/hardImageNet) - is missing to explain how to load the dataset or how to use the evaluation metrics. As it is, I don't know how to use the dataset once I have downloaded it or use the metrics.

**Ethics:**

Amazon Mechanical Turk workers made the annotation. Their working conditions may fall within the scope of the paper's ethical considerations.

**Relation To Prior Work:**

Yes, the authors discussed how their work differ from previous contributions

**Summary And Contributions:**

The authors propose 1) a dataset consisting of 15 imageNet classes for which they have collected object segmentations and established a ranking based on the strength of spurious signals present as well as 2) an evaluation suite of ablation-, noise-, and saliency-based analyses that exploit the object segmentation to assess the extent to which models rely on spurious features.

The annotated examples were chosen because they are, according to the authors, typical examples of images whose class prediction made by a model will be based on spurious cues and not on the real object.

---

> ### Author Response · Authors · 2022-08-12
> **Clarifying Class Selection; Alternate Angle for Ablation**
>
> We thank the reviewer for their detailed feedback. We address concerns below:
>
> - **Class selection for Hard ImageNet**: We do not claim that Hard ImageNet objects necessarily have the strongest spurious cues in all of ImageNet, nor does Hard ImageNet represent the complete subset of ImageNet with strong spurious cues – such a distinction cannot be well defined. Instead, we sought to investigate a subset which we had a strong inclination (based on Salient ImageNet analysis) to believe that models would rely on spurious cues during classification. Through our diverse evaluation suite, we present extensive empirical evidence that validates our hypothesis that models do indeed rely on non-core features far more for Hard ImageNet objects than others.
> The Salient ImageNet work is currently under review, but a previous paper introducing the framework for annotating neural features as core or spurious for a class has been published in a reputable venue (ICLR). We add section E to the appendix in which we discuss the precise methodology of Salient ImageNet in greater detail, and provide some examples of the visualizations that led to the annotation of important features for Hard ImageNet classes as spurious.
> - **On ablation**: We concur with the reviewer that there is not a perfectly clear desired behavior for classifiers in the face of ablated images. However, we believe the drop in accuracy for Hard ImageNet objects relative to other objects is still informative. Nonetheless, we proposed a modified metric for the ablation experiments: instead of looking globally at accuracy, we inspect prediction confidences for each instance. Moreover, we focus on the drop in prediction confidence when the object is removed. This way, we can take a closer look at what a classifier believes it sees. While a classifier may still correctly predict an image based off background alone, ideally we would see a significant drop in confidence. We present these results in appendix section F. Again, we see confidence drops for Hard ImageNet objects are far lower than those for RIVAL20 objects, and generally, results very closely resemble those obtained by inspecting the accuracies under ablation as opposed to prediction confidence.
> While this measure may be imperfect, we find that it is an intuitive measure that may be more easily interpretable than our noise or saliency based metrics. We hope our modification helps address your concern: while it may be unreasonable to expect a model to classify a foreground-ablated image to any other class, it is perhaps more reasonable to expect a model to classify the image to its true class with lower confidence.
> As to the specific example you mention, ideally a model would classify the keyboard with the space bar missing either as ‘Computer Keyboard’ (class index 508) or ‘Typewriter Keyboard’ (class index 878). However, we agree that it is feasible that an ablated image is still most likely to belong to the original class, particularly when similar classes are considered (i.e. in the finetuned setting). Nonetheless, we still find ablation an intuitive and insightful metric, and one with precedent (see only-background splits of ImageNet 9).
> - **User Guide for Dataset**: We add instructions to the github repository for both downloading and setting up the dataset, as well as how to use the metrics, as requested.
>
> We hope our comments sufficiently answer any questions. If so, we would greatly appreciate any increase in score or confidence. Thank you.

---

> > ### Comment · Reviewer_M3X2 · 2022-08-29
> > **Thank you for your answers**
> >
> > Dear Authors,
> >
> > Thank you for taking the time to respond to my comments.
> >
> > - **Class selection for Hard ImageNet:** I thank you for the clarification and appreciate your position better. I just want to point out that I did not find the new section E in the supplemental materials.
> > - **On ablation:** I thank you for detailing what motivates your interest in this metric and appreciate the nuances brought to the metric. Would you consider modifying your paper a bit to clarify these nuances? Also, let me come back to the example of the space bar, you mention in your answer "ideally a model would classify the keyboard with the space bar missing either as 'Computer Keyboard' (class index 508) or 'Typewriter Keyboard' (class index 878). " I also want to add on the example of the space bar that I think that in this case, the mask may not bring a huge difference to the initial image in the sense that the shape of the mask is also informative (which is preserved in the "object grayed" or "bbox grayed" setting for this example) even if I concede that the "texture" and the color are lost and that it would be expected a decrease in confidence. In sum I continue to think that the end of the sentence "Hard ImageNet images are predicted to their class over 50% of the time under all ablations, despite the fact that the object essential to the class label is actually missing, *rendering the predictions incorrect*" present in your paper gives a too quick conclusion.
> > - **User Guide for Dataset:**  Thank you for making these changes. Nevertheless, I would like to point out that I am personally unable to produce your dataset since I do not have an ImageNet-1k archive stored in its original format whereas it seems to me that it is this format that is expected in line 10 - `_IMAGENET_ROOT = '/scratch1/shared/datasets/ILSVRC2012/'` - of the `hard_imagenet.py` file. Unless I'm mistaken, it seems to me that the [https://image-net.org/download.php](https://image-net.org/download.php) site doesn't deliver any ImageNet-1k archive since a while and that the only way to access ImageNet-1k to my knowledge is to use the version available on the HuggingFace Hub at [https://huggingface.co/datasets/imagenet-1k](https://huggingface.co/datasets/imagenet-1k). However the format of the data on this Hub does not correspond to the original format which would require non-negligible work from the person who would like to use your dataset.

---

> > > ### Author Response · Authors · 2022-08-29
> > > **Thank you! Additional updates made to paper**
> > >
> > > Thank you for reviewing our changes and once again for the additional detailed feedback.
> > >
> > > - **Appendix E**: Apologies, we had updated the main pdf to contain the new appendix for ease of viewing. We have now updated both the main text pdf and the supplmental materials so that they include all added content, including the detailed overview of Salient ImageNet.
> > > - **Ablation**: Thank you again for bringing attention to the nuance needed in appealing to this metric. We have changed the wording in the sentence you mentioned in the main text, as well as added reference to a longer discussion in appendix F. We agree that in the case of the space bar, ablating the object does not seem to remove much information at all, seeing as there is very little color, shape, and texture information associated with the object in the first place. We argue that perhaps this simple quality to the object is precisely what makes it so hard to classify faithfully -- it is possible that for certain objects, using context is required to effectively make predictions. We hope that our paper will bring light to how class matters with respect to *how* classifiers operate, leading to greater efforts in making deep classifiers less opaque and more reliable.
> > > - **Data access**: We appreciate you bringing this to our attention. We have added public access to the subset of ImageNet images that our annotations pertain to, as well as the subset corresponding to the baseline in our study (RIVAL20). Specifically, we host the images from the 15 and 20 ImageNet synsets respectively on box, and provide links for easy downloading in case users do not already have ImageNet saved. Upon downloading, one would simply replace the _IMAGENET_PATH field with the path to the downloaded subsets, and the code would work exactly the same as if all of ImageNet was there.
> > >
> > > We truly appreciate all the feedback you've provided, and hope we have been able to sufficiently improve our paper so to satisfy your concerns and maximize the contribution of this work to the community.

---

> > > > ### Comment · Reviewer_M3X2 · 2022-09-01
> > > > **Thank you very much for your answer**
> > > >
> > > > Dear authors,
> > > >
> > > > Thank you again for your answers and additions. I also think that these changes increase the quality of the paper and the proposed dataset.
> > > >
> > > > Best

---

### Official Review · Reviewer_NpX5 · 2022-07-27
**Important contribution and good paper**

**Rating:** 7
**Confidence:** 3
**Clarity:** The writing of this paper is excellen…

**Strengths:**

- The proposed dataset provides a curation of class samples with strong spurious cues selected using up-to-date empirical methods, thus relevant in the current literature.
- Hand-annotated foreground segmentation is provided which can serve as an important tool for analysis and method development regarding spurious cues.
- Extensive documentation on the dataset is provided and the soundness of the data collection procedure is very convincing.
- The quality standards for the data collection procedure is very high and stringent measures were employed to enforce these standards, such as qualification exams and intermediate attention checks.
- The experimental results using the curated evaluation suite are very consistent and clearly support the need for the proposed dataset.
- The paper is very well-written.

**Weaknesses:**

- Justification for using the method from [28] for ranking spurious cues is not provided in relation to alternative methods. Is the method used in the paper the best choice compared to other methods?
- In line 139, the authors state that they *compile* suite of evaluation metrics, suggesting that the metrics have been proposed in previous works. However, there is no citation provided for the Ablation method in Section 4.1.

**Additional Feedback:**

- Typo in Line 190
- Writing is *mildly* hard to follow in lines 83-85 (indirect connection between l2 adversarial training and spurious neural feature) and line 86 (warrants clarification that vast majority of class-features pairs were deemed core *in general*).
- The justification for the use of the method from [28] to measure the spuriousness of the dataset could be tested by comparing the class ranking derived from [28] to a class ranking derived from the proposed evaluation metrics.
- Discussion regarding existing object detection datasets containing object segmentation annotations may be relevant to the scope of the paper. Possible avenues include:
  - Comparison with HardImageNet segmentations for improving models as in Appendix C. I.e., could segmentations from typical non-spurious datasets achieve the same level of spurious feature reliance?
  - Use of additional datasets for justification of the spurious class-ranking method based on [28], in the aforementioned manner (third bullet point).

- It was pleasantly surprising to see the care devoted to the working environment of data annotators. Not only does this improve the well-being of workers, but such a transparent and open environment would arguably make a crucial difference in the quality of the resulting data.

**Correctness:**

All claims appear to be correct.

The construction of the dataset is thoroughly documented in Section 3 and Appendix D and is convincingly sound. The authors impose a high standard of data quality for their collection procedure. There are no noticeable issues in the experiment design.

**Documentation:**

Details of the proposed dataset are thoroughly documented in Section 3 and Appendix D. Particularly, in Appendix D, the authors employ the *Datasheets for Datasets* protocol for detailed documentation.

**Relation To Prior Work:**

The paper provides thorough discussion of related works, and motivation for the proposed dataset as well as differences with previous contributions are clear.

**Summary And Contributions:**

The paper proposes a novel dataset, Hard ImageNet, which is comprised of a curation of 15 classes from ImageNet with very strong spurious cues, as well as their segmentation masks. The dataset is the first of its kind to provide samples containing natural spurious cues in general image domains. The classes are selected using an up-to-date automated saliency method and annotations are carefully collected based on a stringent protocol. The paper also provides a curated suite of evaluations methods for spurious cues in datasets where segmentation masks are available. Experimental results strongly suggest that the proposed dataset contains a high level of spurious cues compared to a typical image dataset counterpart–RIVAL10. Furthermore, the authors present a clear case of how Hard ImageNet can be used to train models to be more robust to spurious cues, going as far as to present experimental results in the Appendix.

---

> ### Author Response · Authors · 2022-08-12
> **Thank you for all the feedback!**
>
> We thank you for your detailed feedback and kind words. We address concerns below.
>
> - **Clarifying relation to Salient ImageNet**: We have added a section in the appendix to clarify the procedure employed in the construction of Salient ImageNet to annotate neural features. We also revise the specific locations in the text you mention; thank you for the precise critique. As mentioned in the updated main text, we were referring to all 5000 class feature pairs (5 important features by class) when we stated that the vast majority (~86%) of the class-feature pairs were core.
> - **Validating Image Rankings**: To our knowledge, our work is the first to attempt to rank images based on the strength of spurious cues present. Thus, we are not aware of alternate methods for performing this task. Our approach was to leverage machine perspective via the activations of annotated neural features. We attempted to validate our selections qualitatively by visualizing the highest and least ranking images for a few classes in Fig 10. The use case (see deep feature reweighting in the appendix) we had in mind for the image rankings only uses images with extreme ranks (highest or lowest), which motivated our brief qualitative validation. This qualitative validation holds as we view a larger number of the highest and lowest ranked images, though we do not include these full visualizations in the paper due to space constraints. We concur that more methods for ranking images would be very useful, both for validating our method and for opening the door to new ways in how we handle data.
> - **On Object Detection Datasets**: Typically, object detection annotations come in the form of bounding boxes. Many of the objects in Hard ImageNet do not lend themselves to being captured in a single box. Thus, we believe the segmentation masks we collect may be of more use for precisely localizing objects (and consequently distinguishing them from surrounding spurious cues).
> - **Citation for Ablation Methods**: We note that we cite a related work in lines 60-62, mentioning that it inspires our ablation analyses. We have added a citation again in section 4.1 for completeness.
>
> Once again, we thank the reviewer for all the feedback. We also appreciate the emphasis placed on fair treatment of data annotators – we made a considered effort to this end and are happy that it was highlighted, as we hope all data annotators are treated fairly.

---

> > ### Comment · Reviewer_NpX5 · 2022-08-24
> > **Thank you for the response**
> >
> > Thank you for taking my comments into consideration and addressing them individually. My concerns have been addressed.
> >
> > Thank you for the detailed clarification of Salient ImageNet in Appendix E; the expalantion is extensive and very clear. I think the qualitative image ranking and DFR ablation are adequately convincing. I agree with your point that segmentation masks are significantly more informative than bounding boxes. I think the additional citation in 4.1 will be helpful for eager readers.
> >
> > ---
> >
> > Best of luck.

---

### Official Review · Reviewer_7A1M · 2022-07-28
**An interesting observation for models to focus on less-centred and significantly smaller objects**

**Rating:** 5
**Confidence:** 4

**Strengths:**

- This data can help the classification models focus on even smaller objects in the image that are hardly distinguishably visible with a large amount of background.
- It can lead to robust models in future research without being much affected by extensive backgrounds and focus precisely on objects to be classified.


**Weaknesses:**

- The paper shows baselining on very few models. Have authors tried other models also which were trained on Imagenet, like versions of VGG and Densenet? Do they also do a similar thing, like not focusing on an object, as shown in this paper?
- No segmentation maps for other classes, just 15 classes of 1000 classes is a tiny chunk of a dataset compared to ImageNet. Do other class samples like kite, snake and many more also deviate from focus, is this checked?


- If we use pre-trained segmentation models, then it can generate segmentation masks like this paper can be used:
                       “Segmentation propagation in ImageNet”, D Kuettel, M Guillaumin, V Ferrari, ECCV-2012.
- No Conclusion in the paper.
- Paper just mentioned about ObjectNet at starting, but no results were shown for ObjectNet images.


**Additional Feedback:**

- Conduct more experiments on other models trained on ImageNet to show baselining results.
- Explore segmentation models for annotating masks.
- Write Conclusion in the paper.
- Show experimental results on the ObjectNet dataset also.

**Clarity:**

- The paper is written well and clearly. The claims made are appropriate.

**Correctness:**

- Interesting observations were noted in the models trained on the ImageNet dataset and a good attempt to create a segmentation mask for objects in the image to deal with the problem of not focussing on smaller and less centred objects.
- Claims made in the paper seem correct; the proof was given by showing saliency maps indicating the model's focus in images.


**Documentation:**

- Sufficient details are present in the paper for the dataset collection procedure.

**Ethics:**

- Dataset is ethically created and uses some classes of ImageNet dataset.
- License is provided, but IRB approval is missing.


**Relation To Prior Work:**


 - It is mentioned in the paper how 15 classes show less-centred and very small objects with extensive backgrounds in the ImageNet dataset.
- Segmentation models have not been explored to create segmentation masks for the ImageNet dataset classes.


**Summary And Contributions:**

- The authors have found fascinating observations of 15 classes in the ImageNet dataset (namely - Dog Sled, Howler Monkey, Seat Belt, Ski, Sunglasses, Swimming Cap, Balance Beam, Gymnastic, Horizontal Bar, Patio, Hockey Puck, Miniskirt, Keyboard Space Bar, Volleyball, Baseball Player, and Snorkel) that rely on spurious features which lead to a decrease in generalization of the model. These objects are generally less centred and smaller as they contain much background in the image. For this, the authors introduced segmentation masks for these images of Imagenet, which are hard-imagenet objects to be recognized by the model.

- They compared Finetuned and pre-trained resnet50 and vision transformer (small DeiT) model for Hard Imaenet objects, RIVAL10, RIVAL20 by taking cases for input image to the model like the original image, grayed object so that it hides the object in the image to be predicted, bbox grayed by creating gray bounding box to even don’t let the model predict through the shape of the object, and bounding box tiled to retain as much background as possible. Also, the saliency maps shown in the paper indicate that the model focuses on background regions, not the object.

- They annotate each image in the training set with a class-wise rank corresponding to the strength of spurious features present in the image based on the neural features of each class. Samples are classified with low accuracy if their spurious rank is low.

---

> ### Author Response · Authors · 2022-08-12
> **New baselines and Clarification on existing Segmentations; Conclusion to come in extra page if paper is accepted**
>
> We thank you for your detailed comments. We address each concern below.
> - **More baselines**: In Appendix D, we additionally evaluate pretrained Swin Transformer, ConViT, DenseNet161, and VGG16,  demonstrating the ease of use of our benchmark. For all models studied, spurious cues are relied upon much more for Hard ImageNet objects than RIVAL20 objects, as expected.
> - **Segmentations for other ImageNet Classes**: All other ImageNet classes have soft segmentation masks provided in the Salient ImageNet dataset. Crucially, those classes had at least one important (i.e. highly contributing to class logit) neural feature that was annotated as core. Thus, soft segmentation masks could be obtained by inspecting Neural Activation Maps of the core feature(s). We refer the reviewer to salient-imagenet.cs.umd.edu to peruse the annotated features for all ImageNet classes, or to download all the soft segmentation masks. While other segmentation models may have been used for this purpose, the goal of our paper is not to generate masks for all objects (as Salient ImageNet already mostly achieves this task). Instead, our masks are intended to focus on an interesting subset, while also completing Salient ImageNet’s aim of collecting object segmentations for all of ImageNet.
> We note that we do not claim that spurious cues are not present in other classes, nor that the spurious cues in Hard ImageNet images are stronger than all other cues. Nonetheless, thanks to the Salient ImageNet analysis, we had good reason to believe the spurious cues in Hard ImageNet objects were stronger than normal, which we then empirically show over various metrics.
> - **ObjectNet results**: We note that over 98% of the classes in ObjectNet do not overlap with Hard ImageNet. Furthermore, ObjectNet images do not have segmentation masks, so our methods for analysis cannot be applied. This highlights a potentially advantageous property of our dataset compared to others that seek to measure distributional robustness. Since we go beyond single class label annotations, we believe that many novel methods for analysis and model training can be developed on Hard ImageNet. As for training more robust models, we believe this out of scope of this paper, as we instead focus on presenting a dataset and associated benchmark. However, we hope the community can use Hard ImageNet to arrive at new methods for more robust training, which could then be validated by checking performance on challenge datasets like ObjectNet.
> - **Conclusion**: We have added a conclusion in the appendix of our revised submission. We will move the conclusion to the main text of the paper if it is accepted, upon which an extra page of content will be made available.
>
> We hope this rebuttal is satisfactory. If you believe our experiments on new models and clarification regarding the existence of segmentation masks for the remainder of ImageNet (via Salient ImageNet) address your concerns, we would greatly appreciate an increase in score, as you are currently the sole reviewer who is not recommending acceptance. Thank you! We are happy to answer any more questions.

---

> > ### Comment · Reviewer_7A1M · 2022-08-20
> > **Thanks for addressing the comments.**
> >
> > Yes, I agree that masks are created intended to focus on an interesting subset, to focus on hard classes of Imagenet. But this seems to be the before step/preprocessing step or some small part which can be used to help models to focus on the specified parts of the image but not enough contribution to the dataset papers.
> >
> > Best of luck !!

---

> > > ### Author Response · Authors · 2022-08-23
> > > **Thank you, our rich annotations can be very useful in training improved models with novel supervision, hence the dataset contribution**
> > >
> > > The annotations we provide go beyond what is used in prevalent classification algorithms. That is, the algorithms for which gathering our segmentations and rankings would be a first step do not exist yet to our knowledge. Furthermore, we’d argue that existing algorithms for segmentation may struggle significantly for the classes in Hard ImageNet, as the segmentation models/methods will likely also be plagued by the strong spurious cues and small non-centered nature of the objects.
> > >
> > > Our dataset will allow for the construction of fundamentally new methods that go beyond single class-label supervision for classifying objects. We explore some preliminary methods to leverage the object segmentations and image rankings for improved classification in appendix C. Seeing as this paper is in the datasets and benchmarks track, we focus our paper more on the data and the diverse benchmark we create for it, as opposed to designing algorithmic solutions. Nonetheless, we note that our paper enables deeper study of the spurious correlations problem in natural data, and also opens the door to brand new approaches for classification.
> > >
> > > Lastly, thank you for acknowledging our additional experiments and clarifications in response to the initial critiques. We hope our rebuttal sufficiently addressed those concerns.

---

### Official Review · Reviewer_RHtp · 2022-07-28
**Very interesting topic with potential impact on many domains. The paper's clarity is not high.**

**Rating:** 6
**Confidence:** 3
**Correctness:** Solid

**Strengths:**

I believe that this work would help understand neural nets better and I find the concept very interesting and impactful for many domains at this field.

The paper illustrates the phenomena and methods with good visualizations.

The comparison of top spurious feature presence vs. non-presence on ImageNet shown in Figure[10] is strong. I'd highlight it, perhaps a teaser image could be derived from it.

The project website along the dedicated GitHub repo provides great support to reproducible science.

**Weaknesses:**

To me the paper was hard to follow. I believe that with some text editing, without changing the content, it would be much easier to read.
It begins with the introduction where a definition in simple words would help any reader understand the challenge the authors try to solve better. In terms  of storytelling I'd try to lead the reader from motivation in each chapter to the details. Some sentences where unnecessarily complex.

**Additional Feedback:**

Other than the class-labels 'Horizontal bar' and 'Balance beam', no two classes share background visual features. Taking multiple objects from the same habitat would inherently minimize the spurious cue factor. It might be interesting to quantify and illustrate on a graph. Y axis would be one evaluation metric while X would be the number of objects that share the same background in the training data.

**Clarity:**

There's room for improvement in the clarity of this manuscript. As mentioned in the weaknesses, I found it hard to follow due to over complication of sentences and non-intuitive paper structuring.
The storytelling in this paper is sub-optimal which could be easily improved as the content of the paper is very interesting.

**Documentation:**

The collection and labeling processes are detailed and well documented.

**Ethics:**

No ethics concerns as this work is derived from existing public datasets.

**Relation To Prior Work:**

Looks ok.

**Summary And Contributions:**

Hard-ImageNet is a dataset with a formulized a task and metrics aimed towards solving an image recognition model unwanted behavior referred to as reliance on spurious cues. This phenomena happens when class labels, in a classification problem, happen to have significant correlation with background features, lightning, textures, and other characteristics. The problem arises when the object appears elsewhere on test/production inference data and that's where the generalization degrades.

The metrics suggested here leverage noise, ablation, and saliency methods to quantify model's resiliency to spurious cues on a given dataset.

Demonstration on convolutional neural nets as well as transformers is provided.

Analysis on spatial localization of such class labels vs. other ImageNet objects.

---

> ### Author Response · Authors · 2022-08-12
> **Thanks for feedback, any extra notes to improve clarity are appreciated**
>
> Thank you for your comments. We have made revisions to the text of the paper to improve clarity, specifically focusing on reducing longer sentences in the introduction and review of literature. If there are any other specific areas you believe we should amend, we would appreciate the feedback. We also added a conclusion* and an appendix (Section E) discussing the Salient ImageNet dataset and the annotations of spurious features for Hard ImageNet classes, as those annotations are what motivated the creation of Hard ImageNet. We hope that we clearly communicated the overall problem of spurious feature reliance, the data-dependency of this problem, and its importance. Further, we note that other reviewers found the text sufficiently clear, describing it as “well written” (7A1M,  M3X2), “easy to follow” (NpX5, TYq1), and “excellent” (NpX5).
>
> We also appreciate the experiment idea you suggest. We highlight that knowledge of the spurious features at play are instrumental in order to carry out this experiment (i.e. we need to know what the spurious cues are for each class in order to group classes by shared spurious cues). While carrying out this experiment in full may be out of scope for this paper, we believe the annotations of Hard ImageNet and Salient ImageNet together may facilitate the experiment, since we can group classes specifically by the neural features that are spurious and important for multiple classes.
>
> To demonstrate this, we carry out a preliminary version of the suggested experiment. Namely, for each neural feature that is spurious and important for a class, we check the number of classes that also rely (i.e. among top-5 most important features) on that neural feature in a spurious way. For neural features relied on by Hard ImageNet objects, the average number of classes that also rely on those features is 2.89. For all other ImageNet objects, this number is 4.10, more than 40% higher. This means that indeed the spurious cues for Hard ImageNet objects are shared by fewer other objects, which may lead the model to rely on them more heavily, as the predictive power of these spurious features is stronger (since they are more unique to the class).
>
> We hope that this additional experiment further demonstrates the utility of our dataset. Furthermore, we hope our revisions have improved the clarity of the text (and we are open to making more changes if you have specific suggestions). If you concur that our dataset is a valuable contribution, any increase in score would be greatly appreciated.
>
> *The conclusion is currently in the Appendix. We will add this to the main text if the paper is accepted, upon which an extra page of content is made available.

---

### Review · Ethics_Reviewer_NsUa · 2022-08-22

**Recommendation:** 1

**Ethics Review:**

The NeurIPS Ethics Guidelines note that
> If the research uses human-derived data, consider whether that data might:

> ...

> 4. Contain human subject experimentation and whether it has been reviewed and approved by a relevant oversight board. For instance, studies predicting characteristics (e.g., health status) from human data (e.g., contacts with people infected by COVID-19) are expected to have their studies reviewed by an ethical board.

The object segmentations were gathered via Amazon Mechanical Turk. I.e., this submission contains a human-subject experiment. Based on the description of the experiment, I did not identify any serious ethical issues with the experimental procedure. However, the experiment is missing an IRB approval, as reported in the submission:

> We did not consult our institutions IRB directly, as our collection procedure closely followed a previously approved protocol.

If the paper had an IRB approval, my ethical recommendation would have been "No serious ethical issues", since I do not see any issues with the experimental procedure itself. However, based on the NeurIPS Ethics Guidelines, it is not clear to me if a lack of an IRB approval would immediately lead to rejection for web-based experiments that do not gather personally identifiable information, such as this one.

---

Update after author response: I would like to thank the authors for contacting their IRB to confirm that their study is exempt from approval. Since this issue is resolved, I have updated my score.

---

> ### Author Response · Authors · 2022-08-23
> **Confirmed with IRB that our study is exempt from approval because it does not constitute human subject research**
>
> tldr: We have again consulted our institution's IRB and confirmed that IRB approval is and was not necessary for our study, as our data annotators are not human subjects.
>
> long:
> Hello, today we received explicit confirmation from Joseph Smith, the director of the Human Research Protection Program in the Division of Research at University of Maryland that our work does not require IRB approval. We followed IRB protocols in making our inquiry and Joseph Smith speaks on behalf of the IRB when he says because "you are not asking about the individual, then this project does not meet the definition of human subject research and IRB Approval is not required".
>
> We assure that we strived to maintain the highest standard of ethical considerations during our research. If there is anything we can do to further remedy concerns, we will happily do so. We now provide some explanation as to why our procedures do not constitute human subject research as defined by law. The reference material for this explanation comes from the CITI training course "Defining Research with Human Subjects" (SBE ID 491).
>
> IRB approval is required for *human subject research*. When the definition of *human subject* or *research* is not met, then IRB approval is not required. A *human subject* is defined as follows:
>
> >According to the federal regulations at 45 CFR 46.102 (Protection of Human Subjects 2018), a human subject is a "living individual *about whom* an investigator (whether professional or student) conducting research: (i) obtains information or biospecimens through intervention or interaction with the individual, and uses, studies, or analyzes the information or biospecimens; or (ii) obtains, uses, studies, analyzes, or generates identifiable private information or identifiable biospecimens.
>
> The 'about whom' wording does not apply in our case, as Joseph Smith confirmed. As explained further,
>
> >Another key part of the human subjects definition is the “about whom” wording. Some research that involves interactions with living individuals does not meet the regulatory definition of research with human subjects because the focus of the investigation is not on the opinions, characteristics, or behavior of the individual. Instead, the individual is asked to provide information about something. How many micro-loans were made last year? What is the average amount of those loans? These are not "about whom" questions, but can be thought of as "about what" questions.
>
> We do not collect any information on the opinions, characteristics, or behavior of the annotators. Instead, we simply ask the annotators to provide information regarding the location of objects in images. Thus, since our research does not involve human subjects as defined above, we are exempt from IRB review:
>
> >A study that meets the federal regulation’s definition of research, but does not involve human subjects, does not need IRB review.
>
> Our research team has recently collected data annotations with near identical procedures used in this work. Namely, we closely resemble the procedures for collecting RIVAL10 annotations, which has been published to CVPR 2022. In that prior experience, we had confirmed that IRB approval was not required. Nonetheless, we attempted to incorporate the knowledge and principles learned from the history of human subject research (i.e. providing informed consent, ensuring the rights of the participants, anonymizing responses, keeping work entirely transparent, voluntary, and justly compensated) in order to uphold the highest ethical standards in our procedures. We hope that this additional external confirmation from our institution's IRB remedies your concerns regarding the ethics of our study.
>
> If you agree, we would greatly appreciate if you adjusted your ethical recommendation accordingly. If you have any additional questions or requests, we would be more than happy to answer.

---

> > ### Author Response · Authors · 2022-08-29
> > **Thank you for the score update! We've made clarifications in paper re IRB approval**
> >
> > Thank you for reading our response and updating your review to "no serious ethical concerns". We have additonally modified the language in appendix G.3 to reflect why our study is IRB exempt and to clarify that we nonetheless confirmed this with our institution's IRB, as well as maintained core ethical principles in our procedures with humans in the loop.

---

### Comment · Reviewer_M3X2 · 2022-07-25
**mmoayeri.github.io/HardImagenet returns a 404 error**

Dear Authors,

The link indicated for the dataset - [mmoayeri.github.io/HardImagenet](mmoayeri.github.io/HardImagenet) - returns me a 404 error. As a reviewer, do we have to do anything special to access it?

Best

---

> ### Author Response · Authors · 2022-07-25
> **Case Sensitive Website Link: https://mmoayeri.github.io/HardImageNet/**
>
> Sincerest apologies for the confusion - the ‘N’ in HardImageNet needs to be capitalized, so the website link is actually mmoayeri.github.io/HardImageNet.
>
> Thank you for bringing this to our attention!

---

### Meta-Review · Area_Chair_1kSq · 2022-09-10

**Recommendation:** Accept
**Confidence:** 4

**Metareview:**

The paper has gone through the ethics review. As per the suggestion of the ethics reviewer, the authors confirmed with the IRB of their institute that IRB approval is not needed for the study. The paper thus does not have serious ethical concerns.

After the discussion period, the paper obtains an average rating of 6.2 from 5 reviewers. Most reviewers like this paper. They find the analysis provided in this paper interesting and novel, highlighting the implications of spurious features in recognition. They also appreciate the new subset of 15 ImageNet classes that are labeled with segmentation masks. The dataset is documented well.

Reviewers such as RHtp and NpX5 asked for an improvement in clarity. The authors improved the paper accordingly. The authors also clarified the selection of the 15 classes and the ablation. Reviewers are happy with the rebuttal and revised content.

Reviewer 7A1M is the only reviewer who does not recommend acceptance (rating = 5) after the discussion period. After checking the rebuttal, the AE believes the authors have addressed the concerns of Reviewer 7A1M, by including the additional experiments and missing conclusion. The current dataset and analysis of the paper are still meaningful without the segmentation annotations on other ImageNet classes and the results on ObjectNet.

---

### Decision · Program_Chairs · 2022-09-16

Accept